# Recent Advances in the Preparation, Structure, and Biological Activities of β-Glucan from *Ganoderma* Species: A Review

**DOI:** 10.3390/foods12152975

**Published:** 2023-08-07

**Authors:** Henan Zhang, Jingsong Zhang, Yanfang Liu, Chuanhong Tang

**Affiliations:** Institute of Edible Fungi, Shanghai Academy of Agricultural Sciences, National Engineering Research Center of Edible Fungi, Key Laboratory of Edible Fungi Resources and Utilization (South), Ministry of Agriculture, Shanghai 201403, China; henanhaoyun@126.com (H.Z.); syja16@saas.sh.cn (J.Z.); aliu-1980@163.com (Y.L.)

**Keywords:** *Ganoderma*, β-d-glucan, structural characteristics, bioactivity, application

## Abstract

*Ganoderma* has served as a valuable food supplement and medicinal ingredient with outstanding active compounds that are essential for human protection against chronic diseases. Modern pharmacology studies have proven that *Ganoderma* β-d-glucan exhibits versatile biological activities, such as immunomodulatory, antitumor, antioxidant, and antiviral properties, as well as gut microbiota regulation. As a promising polysaccharide, β-d-glucan is widely used in the prevention and treatment of various diseases. In recent years, the extraction, purification, structural characterization, and pharmacological activities of polysaccharides from the fruiting bodies, mycelia, spores, and fermentation broth of *Ganoderma* species have received wide attention from scholars globally. Unfortunately, comprehensive studies on the preparation, structure and bioactivity, toxicology, and utilization of β-d-glucans from *Ganoderma* species still need to be further explored, which may result in limitations in future sustainable industrial applications of β-d-glucans. Thus, this review summarizes the research progress in recent years on the physicochemical properties, structural characteristics, and bioactivity mechanisms of *Ganoderma* β-d-glucan, as well as its toxicological assessment and applications. This review is intended to provide a theoretical basis and reference for the development and application of β-d-glucan in the fields of pharmaceuticals, functional foods, and cosmetics.

## 1. Introduction

*Ganoderma* has been one of the most popular mushrooms, both for medicine and food, in Asia for more than 2000 years [1]. Currently, approximately 300 species of *Ganoderma* have been identified worldwide [2], which principally includes *G. lucidum*, *G. applanatum*, *G. tsugae*, *G. boniense*, *G. atrum*, and *G. sinense*, and these *Ganoderma* species have a wide range of health-promoting effects. Recently, consumers have become more health conscious, resulting in a growing demand for high-quality, nutritious, and health-promoting natural products. *Ganoderma* is known as a “herb that brings the dead back to life” and has been used for centuries as a medicine or nutritional supplement for the prevention, control, and treatment of chronic diseases [3], such as autoimmune diseases, cardiovascular diseases, diabetes, digestive diseases, and malignant tumors. As a natural medicinal mushroom, *Ganoderma* is abundant in a variety of physiologically bioactive components (more than 600 compounds), including polysaccharides, steroids, triterpenes, glycoproteins, sterols, peptides [4], etc. In particular, polysaccharides are the principal bioactive ingredients found in the fruiting bodies, mycelia, spores, and fermentation broth of *Ganoderma* species; thus, they are considered as an important candidate for functional food, medicine, and cosmetics [5,6].

Among these *Ganoderma* polysaccharides, β-d-glucan acts as the main active polysaccharide and has received increasing attention as a result of its versatile pharmacological activities over the last decades [7], such as its antitumor, anti-inflammatory, antioxidant, immunomodulatory properties. In addition, β-d-glucan exhibits diverse biological and health-promoting effects depending on its natural sources (e.g., yeast, mushroom, bacteria, seaweed, and cereal) [8,9], extraction and purification strategies [10], and structural characteristics [11]. Apart from its pharmacological and nutritional values, β-d-glucan is also used in the food industry due to its gel-forming and thickening effects. For example, the addition of β-d-glucan contributes to improving the texture and sensorial properties of chicken breast [12], sausage [13], and ice cream [14]. Furthermore, β-d-glucan is widely applied in cosmetics industries for its ability to maintain skin health [8,15]. Therefore, β-d-glucan might be regarded as a potential therapeutic agent for diseases, a healthy dairy food, and for use in skin care supplements.

Over the past two decades, the extraction and purification, structural features, bioactivities, and probable mechanisms of β-d-glucan obtained from diverse mushrooms have all been extensively studied by our team and other researchers; however, few reviews have generalized and summarized β-d-glucan from the *Ganoderma* species, which seriously limits the development and utilization of *Ganoderma* polysaccharides. This review aimed to systematically summarize the research advances in the physicochemical properties, structural characteristics, bioactivities, potential action mechanisms, safety assessment, and applications of β-d-glucan derived from *Ganoderma* species, to provide a scientific foundation for the thorough development and use of β-d-glucan from mushrooms.

## 2. Extraction, Purification, and Structural Characteristics of β-d-Glucan

### 2.1. Extraction and Purification

Currently, polysaccharides (especially β-d-glucan) are one of the major bioactive macromolecules components in *Ganoderma* species, and hot water (HW) and chemical solutions (alkali or acid-alkali) are the most commonly used for polysaccharides extraction [16,17]. Meanwhile, the yield of β-d-glucan from *Ganoderma* species using traditional extraction methods (especially HW extraction) may be influenced by some extraction parameters [18], such as the extraction time, extraction temperature, and water-to-raw material ratio. At present, several assisted extraction strategies, including physical methods (microwave, ultrasound, and pressure), biological methods (enzyme), and combinations of these methods (ultrasonic/microwave/enzyme-assisted extraction), have been developed to overcome some of the drawbacks of traditional extraction methods for mushroom polysaccharides [19,20]. Alzorqi et al. [21] showed that the total content of β-d-glucan extracted by ultrasonic-assisted extraction (UAE) from the fruiting bodies of *G. lucidum* was higher than that of enzyme-assisted extraction (EAE) or HW extraction. Smiderle et al. [22] found that the extraction of β-d-glucan from *G. lucidum* using microwave-assisted extraction (MAE) could reduce the extraction time compared with pressurized liquid extraction (PLE). It is worth noting that the above-mentioned extraction techniques help to increase the extraction yield of β-d-glucan from *Ganoderma* species and improve the pharmacological activities of β-d-glucan [23]. After extraction, the crude polysaccharides from *Ganoderma* species were separated and purified by a series of purifications process including deproteinization and decolorization to obtain homogeneous fractions. Subsequently, column chromatography (i.e., cellulose chromatography, ion-exchange chromatography, and gel-filtration chromatography) was commonly employed to further purify the crude polysaccharides [24,25]. Finally, the purified polysaccharides were concentrated, dialyzed, and lyophilized to facilitate the subsequent structural characterization. Of note, the high purity of β-d-glucan from *Ganoderma* species was usually precipitated with absolute ethanol several times. Based on the published literature, the representation of extraction and purification methods for β-d-glucan is summarized and displayed in Table 1 and Figure 1.

### 2.2. Structural Characteristics

Up until now, approximately 200 polysaccharides have been extracted and identified from the fruiting bodies, mycelia, spores, and fermentation broth of *Ganoderma* species [1]. Intriguingly, β-d-glucan was found to be the most promising polysaccharide present in the *Ganoderma* species [27]. According to the published article, high-performance size-exclusion chromatography (HPSEC) and HPSEC-refractive index-multi-angle laser light scattering were performed to determine the average molecular weight (*M_w_*) of polysaccharides (Figure 2A,B), and the *M_w_* of β-d-glucan ranged from 1 × 10^3^ Da to 1 × 10^6^ Da [53]. In addition, Fourier-transform infrared spectroscopy and nuclear magnetic resonance spectroscopy were most commonly used to identify the chemical structure of β-d-glucan. The analysis of the structural features indicated that β-d-glucan from *Ganoderma* species was a liner polymer composed of glucose molecules connected by a β-d-(1→3), -(1→4), and -(1→6)-linked main chain and β-d-(1→6)-linked branches [10,54,55]. For example, a novel β-d-glucan from the fruiting bodies of *G. lucidum* mainly consisted of (1→3)-β-d-glucan with a (1→6)-β-d-glucopyranosyl side-branching unit on every third residue [56]. A water-soluble β-d-glucan from the spores of *G. lucidum* was composed of a mixed (1→3)-, (1→4)-, (1→6)-β-d-glucan backbone with two single β-d-Glc*p* and β-d-Glc*p*-(1→4)-β-d-Glc*p*-1→ disaccharide units in the side chains [46]. Of note, the linkage of the β-d-glucan backbone is the main factor affecting its bioactivity; for example, the antitumor activity of β-d-glucan is related to β-d-1,3-glycosidic [57]. The identified chemical structure of β-d-glucan in *Ganoderma* species is shown in Figure 2C.

## 3. Biological Activities and Molecular Mechanisms of β-d-Glucan

Currently, *Ganoderma* species are used as traditional food and medicine in China and other Asian countries due to their health-promoting effects. In particular, β-d-glucans extracted and purified from *Ganoderma* species have been shown to bring beneficial effects to human health, including immunomodulation, antitumor, antioxidant, and anti-inflammation properties. Moreover, this review summarizes the biological activities of β-d-glucan in Table 2, and its action mechanisms are presented in Figure 3 and Figure 4.

### 3.1. Immunomodulatory Activity

Numerous studies have shown that *Ganoderma* polysaccharides maintain “the stability of the body’s internal environment” by enhancing the immune function of the host [67,68]. Of note, β-d-glucan mainly consists of a repeating structure of _D_-glucose units in *Ganoderma* species, which exhibits an immunomodulatory effect and regulates innate immune functions [69]. Studies have also confirmed that β-d-glucan with a backbone of (1→3)-Glc*p* exhibits a significant immunomodulatory activity [70]. As an immunomodulator, β-d-glucan from *Ganoderma* species can alleviate the development and progression of immune-related diseases by the activation of immune cells (i.e., NK cells [60], dendritic cells [58], lymphocytes [55], and macrophages [71]). For instance, treatment with β-d-glucan from *G. sinense* promoted splenocyte B-cell proliferation and increased inflammatory cytokine secretion in mononuclear cells and DC, as well as enhancing the nitric oxide level in RAW264.7 cells [37]. A randomized clinical trial showed that the administration of β-d-glucan from Lingzhi or Reishi medicinal mushroom enhanced the count of peripheral blood total lymphocytes in asymptomatic children 3 to 5 years old for 7 weeks, and no serious side effects were observed [72]. Moreover, *Ganoderma* β-d-glucan contributes to the protection of spleen and thymus function and increases IgA levels in the serum of cyclophosphamide-induced immunosuppressive model mice [24]. Currently, published articles have indicated that *Ganoderma* β-d-glucan may boost the body’s immune system against COVID-19 [73,74]. Mechanistically, β-d-glucan derived from *Ganoderma* species possesses an inhibitory effect on immunosuppressive diseases through pathogen-associated molecular patterns (PAMPs) (Figure 3). For example, β-d-glucan treatment significantly activated mitogen-activated protein kinases (MAPKs) and nuclear factor-κB (NF-κB) signaling pathways through binding with the pattern recognition receptors (i.e., dectin-1) to induce immune responses [75] and accelerating the secretion of cytokines (i.e., interferon-γ (IFN-γ), tumor necrosis factor-α (TNF-α), and interleukins) by immune cells [76,77]. Other studies found that dectin-1 can cooperate with other specific pattern recognition receptors (i.e., toll-like receptor 2 and complement receptor 3) of β-d-glucan can trigger innate immunity [78,79]. Furthermore, recent studies found that β-d-glucan ameliorated the progression of immunosuppressive diseases via regulation of the gut microbiota composition [80].

### 3.2. Anti-Inflammation

Inflammation is a comprehensive self-protective response of the body’s defense against infection, pathogens, traumas, allergens, and irritants. As anticipated, the anti-inflammatory function of β-d-glucan on humans has recently attracted considerable attention [81]. For example, treatment with a water-soluble β-d-glucan from the *G. lucidum* spores could promote small intestinal crypt epithelial cell proliferation and reduce the levels of pro-inflammatory cytokines, including NO, IL-6, and IL-1β, induced by lipopolysaccharide [6]. Another study confirmed that β-d-glucan from the fruiting bodies of *G. lucidum* is a favorable potential anti-inflammatory agent, which could suppress not only _L_-selectin-mediated inflammation, but also inhibit the proliferation of mouse spleen lymphocyte and human periphery blood lymphocytes [82]. Mechanistically, β-d-glucan showed potential anti-inflammation by blocking the NF-κB pathway [59] and MAPK pathway [7] (Figure 3).

### 3.3. Antitumor Activity

To date, the high morbidity and mortality of malignancies have become a huge challenge for global public health. Meanwhile, the clinical efficacy of traditional treatment methods (i.e., surgery, radiotherapy, chemotherapy, and immunotherapy) is far from satisfactory, and even the resistance of tumor cells to chemotherapeutic drugs can accelerate tumor progression [83,84]. Clinical studies have confirmed that malignant tumors are characterized by the uncontrollable proliferation, migration, and invasion abilities of cancer cells [85]. Encouragingly, β-d-glucan acts as one of the components in *Ganoderma* polysaccharides and it has been shown to have an antitumor activity without side effects [48,86], and so it could be used as a promising therapeutic agent for cancer treatment, which is in demand in clinical applications. For example, the administration of 80 mg/kg β-d-glucan from *G. formosanum* inhibited tumor growth in the lung cancer mice model, and activated the immune response (i.e., enhanced NK cells) and increased the cytokine levels [63]. Another study proved that *G. lucidum*-derived β-d-glucan showed a cytotoxic activity against leukemic cell proliferation and induced cell apoptosis in vitro, and enhanced pro-apoptotic protein (Bax) expression and reduced anti-apoptotic protein (Bcl-2) expression [33]. Moreover, mushroom β-d-glucan containing (1→3)-β-d-glycosidic in the main chain followed by (1→6)-β-d-glycosidic in the side chain is the most effective structural feature with an antitumor activity [87,88]. For instance, (1→3)-β-d-glucan from the fruiting bodies of *G. tsugae* inhibited tumor growth [89]. Functionally, administration with β-d-glucan hampered the progression of colon cancer by increasing the number of beneficial intestinal microbiota [90,91] and the fermented product (i.e., SCFAs). Meanwhile, treatment with β-d-glucan from *G. lucidum* suppressed the malignant biological behavior (i.e., proliferation, invasion, migration, and angiogenesis) of cancer cells via the inhibition of the EGFR/AKT pathway [92] and ERK1/2 pathway [93] (Figure 4).

### 3.4. Antioxidant Activity

Excess reactive oxygen species (ROS) produced during oxidative stress in the human body is a major factor that can cause Alzheimer’s disease, diabetes, atherosclerosis, cancer, and other diseases [94]. Clinical studies have confirmed that the reduction of ROS levels by antioxidants may reduce the risk of chronic diseases and age-related health problems [95,96]. Of note, *Ganoderma* polysaccharides (β-d-glucan) serve as natural antioxidants and have been reported to exhibit a stronger antioxidant activity when scavenging different radicals [25,97]. For example, treatment with β-d-glucan from the fruiting bodies of *G. lucidum* could reduce ROS levels induced by H_2_O_2_, as well as inhibit SMase activity [30]. Another study confirmed that a novel β-d-glucan obtained from the mycelia of *G. capense* had a DPPH radical-scavenging ability and an effective concentration 50 value of 3.23 μM [98]. The above results indicate that the potent antioxidant activity of β-d-glucans from *Ganoderma* species would lay the foundation for their wide application in cosmetic, anti-aging, and pharmaceutical industries. However, the regulatory mechanisms through which β-d-glucan exerts an antioxidant activity are still largely unknown.

### 3.5. Effect of β-d-Glucan on Gut Microbiota

Nowadays, the importance of the gut microbiota in human health and diseases has attracted many researchers’ attention [99]. Recently, alterations in the gut microbiota have been highly correlated with the levels of SARS-CoV-2, as well as the severity of patients with COVID-19 [100]. Over the past decade, numerous researches have demonstrated that oral polysaccharides can interact with the gut microbiota to exert nutritional or pharmacological effects [101]. Mechanistically, polysaccharides can regulate the composition of the gut microbiota and modulate the production of gut microbiota metabolites, resulting in the production of a series of metabolites such as SCFAs, secondary bile acids, tryptophan, and indole derivatives. Of note, several in vitro experiments have shown that β-d-glucan treatment facilitated the growth of *Lactobacilli* and *Bifidobacterial* [102,103]. Similarly, β-d-glucan isolated from the spores of *G. lucidum* ameliorated dextran sodium sulfate-induced colitis by increasing the number of SCFA-producing bacteria and reducing pathogens [48]. Sang et al. [104] reported that β-d-glucan extracted from the sporoderm-broken spore of *G. lucidum* improved high-fat diet-induced obesity, hyperlipidemia, and inflammation through modulation of the gut microbiota composition and enhancing SCFA production. In addition, *Ganoderma* β-d-glucan inhibited the progression of malignant tumors by modulating the composition of the gut microbiota and increasing SCFA production [91,105]. The above results indicate that β-d-glucan from *Ganoderma* spp. has a significant impact on changes in the gut microbiota and in turn on human health.

### 3.6. Others

Besides the biological activities mentioned above, β-d-glucan from *Ganoderma* species exerts other bioactivities, including antimicrobial [42], hepatoprotective [65], anticoagulant [106], antihypertension [107] properties. Meanwhile, Pillai et al. showed [108] that β-d-glucan from *G. lucidum* exhibited a radioprotective activity with a DNA repairing capacity in human lymphocytes exposed to γ-radiation. Of note, with the novel coronavirus pneumonia continuing to spread around the globe, mushroom β-d-glucan has shown a remarkable antiviral ability [109]. Moreover, β-d-glucan administration could alleviate intestinal diseases through the modulation of the intestinal flora homeostasis [48,91].

## 4. Relationships between the Structure and Bioactivity of β-d-Glucan

The versatile biological activities of β-d-glucan are related to its complex structure features (i.e., *M_w_*, monosaccharide compositions, configurations of main and branch chains, and specific glycosidic linkages) [10]. For example, β-(1,3-1,4)-d-glucan or β-(-1,3,6)-d-glucan showed a good antioxidant activity [110]. Furthermore, the triple-helix β-(1,3)-glucan possessed anti-tumor effects through activation of the immune-related pathways by inhibiting the malignant behavior of cancer cells [111]. For example, the triple-helix conformation of β-d-glucan from the fruiting bodies of *G. lucidum* could stimulate lymphocyte proliferation and promote macrophages to form pseudopodia [55]. Moreover, the bioactivities of β-d-glucan were affected by its *M_w_*. For example, a larger-*M_w_* β-d-glucan (1.07 × 10^5^ Da) exhibited a better immunomodulatory activity than low-*M_w_* β-d-glucan (1.95 × 10^4^ Da) [24], which may be related to its direct recognition by specific receptors on the surface of immune cells [112]. Other studies proved that low-*M_w_* β-d-glucan presents various beneficial bioactivities, such as modulation of the gut microbiota [113], as well as anti-inflammation [114], hypoglycemic [115], anti-tumor [116] properties. At present, the relationship between the chemical structures of β-d-glucan and its biological activities has not been fully elucidated, and this mechanism needs to be studied in depth. It has been shown that β-d-glucan with a triple helix conformation and a certain degree of branching can exhibit versatile biological activities [117].

## 5. Safeties

Currently, numerous studies have confirmed the beneficial effects of β-d-glucan on human health, but few studies have analyzed the toxicity and safety of β-d-glucan from *Ganoderma* species. Preclinical studies found that 0.1–10 mg/mL of polysaccharides from the *G. lucidum* mycelia did not delay the hatching and teratogenic defect on Zebrafish embryos at 24 and 120 h post-fertilization [118]. Chen et al. [119] showed that the administration of β-d-glucan (0, 500, 1000, and 2000 mg/kg/day for 90 days) did not cause any toxicologically significant treatment-related changes in clinical observations, ophthalmic examinations, body weights, body weight gains, feed consumption, and organ weights, as well as no cytotoxic effects on the hematology, serum chemistry parameters, urinalysis, or terminal necropsy, which indicated the safety of β-d-glucan application. A clinical study showed that a total of 88 patients with urinary tract infections were treated with *Ganoderma* polysaccharides for 12 weeks, and all of the patients had no signs of liver, hematological, or renal toxicities [120]. Until now, a large number of bioactive β-d-glucans from *Ganoderma* species have been extracted and purified, but the ongoing or completed toxicological studies of these β-d-glucans is just the tip of the iceberg. Therefore, it is necessary to conduct in-depth toxicological studies on β-d-glucan to facilitate confirmation of its efficacy, safety, and potential mechanisms in animal experiments and human trials. Meanwhile, follow-up studies that solve these issues will provide more reliable toxicity and safety data for the application of β-d-glucan from *Ganoderma* species in food, medicine, and cosmetics.

## 6. Applications of β-d-Glucan

As human living standards continue to improve and health awareness grows, consumers are paying more attention to diet and medication for health care. Currently, *Ganoderma* species that are both edible and medicinal and offer health advantages are now growing in popularity among consumers. Pharmacological and clinical studies have confirmed that polysaccharides extracted from the fruiting bodies, mycelia, spores, and fermentation broths of *Ganoderma* species have versatile biological activities such as immunomodulation, antitumor, antioxidant, anti-inflammatory, and anti-aging properties [121,122,123], which are widely used in functional foods, multi-purpose drugs, and cosmetics. For example, several healthcare products and foods containing polysaccharides from the fruiting bodies, mycelia, spores, and fermentation broths of *Ganoderma* species have been developed and produced in markets across the globe, including drinks, healthy wine, jams, and cookies [124,125,126]. Meanwhile, some pharmaceutical commercial products containing *Ganoderma* polysaccharides are used as dietary supplements for humans in the form of powders, oral liquids, and capsules; in particular, capsule products are used as adjuvant drugs for tumor treatment. Of note, β-d-glucan has been widely used in food and pharmaceutical industries due to its physical properties such as water solubility, viscosity, and gelation. For example, Vanegas-Azuero et al. [127] demonstrated that yogurt containing β-d-glucan showed a high percentage of free amino acids, faster protein hydrolysis, better texture parameters, and high sensory acceptability. A study on healthy children found that the administration of yogurt enriched with β-d-glucan from *G. lucidum* increased the frequency of peripheral blood total lymphocytes (CD3^+^, CD4^+^, and CD^8+^ T cells), which are critical elements for the body’s defense against infectious threats [72]. A randomized controlled trial by Vlassopoulou et al. [128] found that the administration of β-d-glucan with a dosage ranging from 2.5–1000 mg/day for 6.5 months significantly enhanced immune defense, improved allergic symptoms, and decreased comorbid conditions associated with obesity. Another clinical trial confirmed that patients with angina pectoris taking 750 mg/day of β-d-glucan for 90 days had increased superoxide dismutase (SOD) levels, decreased malondialdehyde (MDA) concentrations, and reduced numbers of circulating endothelial cells and endothelial progenitor cells [129]. Moreover, *Ganoderma* polysaccharides have broad application prospects in animal husbandry and the feed industry as a green and natural feed additive with rich biological functions [130]. New research has shown that β-d-glucan from *G. lucidum* exhibits whitening effects by reducing tyrosinase activity and melanin synthesis [64]. As a medication delivery system, β-d-glucan has become an important topic in today’s research. For example, Takedatsu et al. [131] created a complex form consisting of macrophage-migration inhibitory factor (MIF) and two single schizophyllan (SPG) chains (β-d-glucan) as a new delivery system for antisense oligonucleotides, and treatment with this antisense MIF/SPG complex effectively inhibited MIF production and reduced intestinal inflammation in a dextran sodium sulfate-induced colitis mice model. Collectively, commercial products of *Ganoderma* β-d-glucan have obtained popularity among humans worldwide for their versatile bioactivities and for being “green and natural”, without side effects.

## 7. Conclusions and Future Perspectives

Over the past half-century, polysaccharides obtained from natural sources have received increasing attention owing to their diverse health benefits. *Ganoderma* is rare medicinal fungi mushroom that has been cultured and consumed worldwide for centuries; it has been used as a traditional remedy for many diseases, including cancer, cardiovascular diseases, allergies, and lung deficiency coughs. Polysaccharides are extracted from various *Ganoderma* species and have the advantages of a low toxicity and broad medicinal value. Because of these properties, *Ganoderma* polysaccharides have been recognized as functional foods and are considered as a source for the development of drugs, nutritional products, and cosmetics. Importantly, *Ganoderma* β-d-glucan serves as an effective and desirable polysaccharide to infuse immune health benefits into any kind of food, dietary supplements, pharmaceuticals, and cosmetics.

Based on the fact that *Ganoderma* species have very different structures and efficacy in different cultured regions, sharing the research results related to *Ganoderma* species from different regions can enable efficient utilization of *Ganoderma* polysaccharides and thus broaden its potential market value. Meanwhile, the bioactivities of β-d-glucan extracted from edible and medicinal *Ganoderma* have received much attention in the biomedical field. However, to further improve the utilization of β-d-glucan in *Ganoderma* species, future research should focus on the following directions: (1) there is an urgent need to select excellent strains, culture techniques, and/or fermentation strategies to improve the shortage of wild *Ganoderma* species; (2) using synthetic biology combined with genetic engineering to enhance the productivity and yield of β-d-glucan in *Ganoderma* species; (3) revealing the relationship between structure and biological activities of β-d-glucan through multi-omics strategies, such as transcriptomics, nutrigenomics, proteomics, and metabolomics; (4) developing food, pharmaceutical, and cosmetic products with β-d-glucan as a functional component and analyzing its safety and toxic side effects; (5) seeking effective physicochemical methods (i.e., ultrasound and microwave) to overcome the special physical properties of *Ganoderma* β-d-glucan (i.e., high *Mw*, linkage pattern and high viscosity) may help to obtain small *M_w_* of β-d-glucan with good absorption and utilization; (6) More clinical studies are required to investigate the use of food-grade β-d-glucan in humans because the majority of pharmacological studies of β-d-glucan are restricted to in vitro or animal models.

In summary, β-d-glucan extracted from *Ganoderma* species will have great market potential for use in food, pharmaceuticals, and cosmetics. In addition, overcoming the above-mentioned drawbacks will provide new ideas for the development of natural β-d-glucans into highly efficient and low-toxic novel products.

## Figures and Tables

**Figure 1 foods-12-02975-f001:**
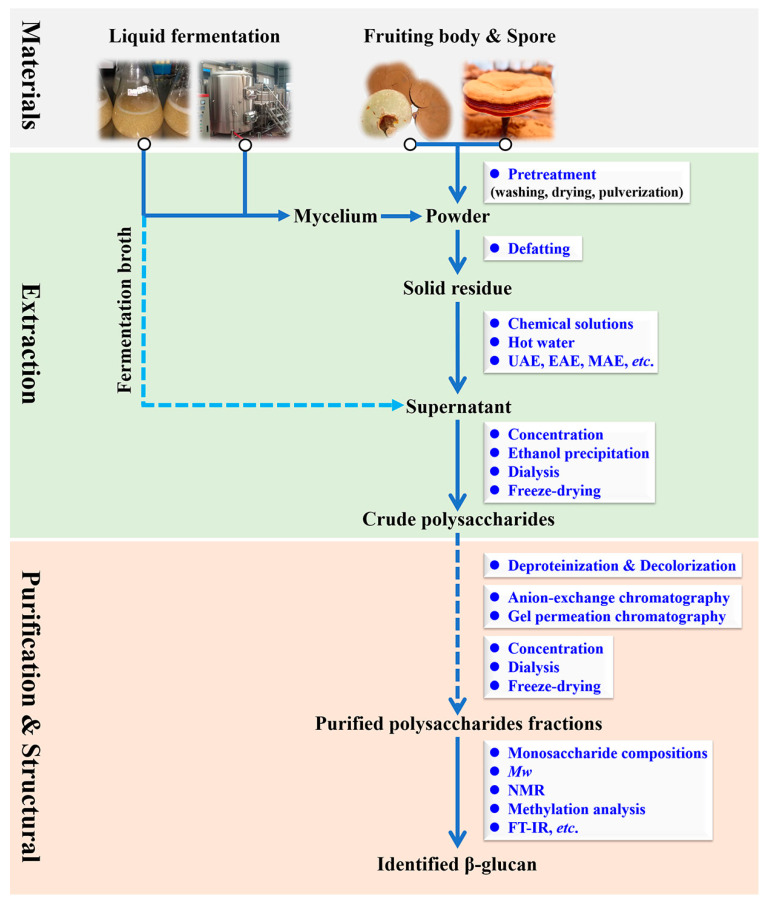
Extraction, purification, and structural characterization of β-d-glucan from *Ganoderma* species.

**Figure 2 foods-12-02975-f002:**
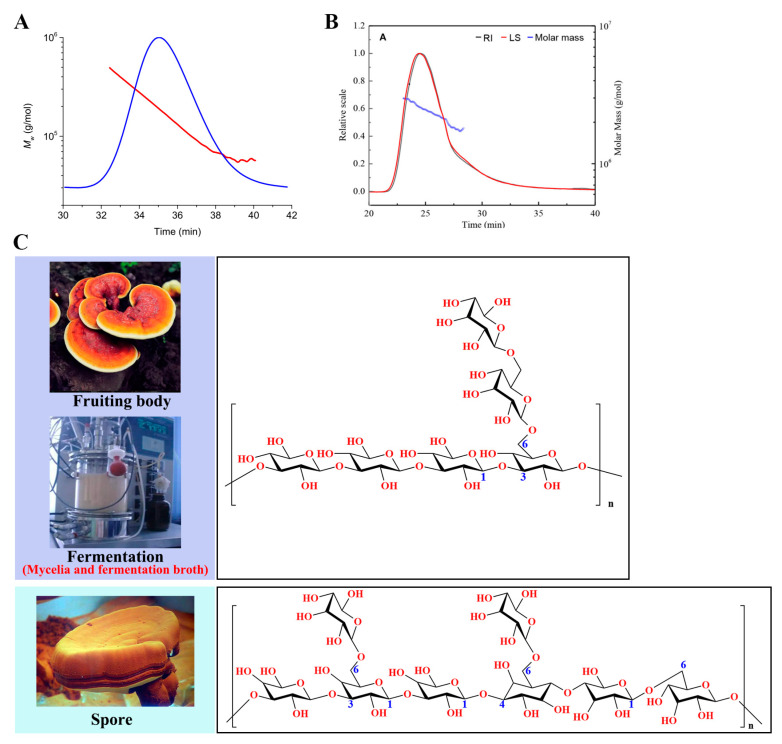
The structural characteristics of β-d-glucan from *Ganoderma*. The HPSEC elution curve and molecular mass distribution of β-d-glucan from *G. lucidum* unbroken spores (**A**) and *G. lucidum* fruit bodies (**B**), Red: differential refractive index; Blue: Molar mass. (**C**) β-d-glucan extracted from the fruiting bodies, mycelia, and fermentation broth of *Ganoderma* mainly consisting of a backbone of β-d-(1,3)-glucan with a side-chain of β-d-(1,6)-glucan, and β-d-glucan extracted from the spores of *Ganoderma* mainly connected by β-d-(1,3-1,4)-glucan as the main chain and β-d-(1,6)-glucan as the side chain linkage.

**Figure 3 foods-12-02975-f003:**
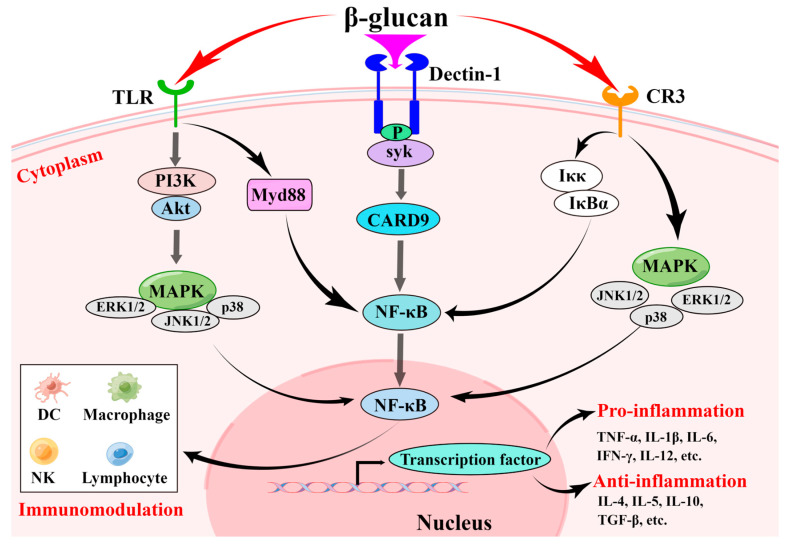
The schematic representation of the proposed mechanisms of the immunomodulatory and anti-inflammatory activity of β-d-glucan. This figure is drawn with Figdraw (www.figdraw.com, accessed on 14 June 2023) (color figure online).

**Figure 4 foods-12-02975-f004:**
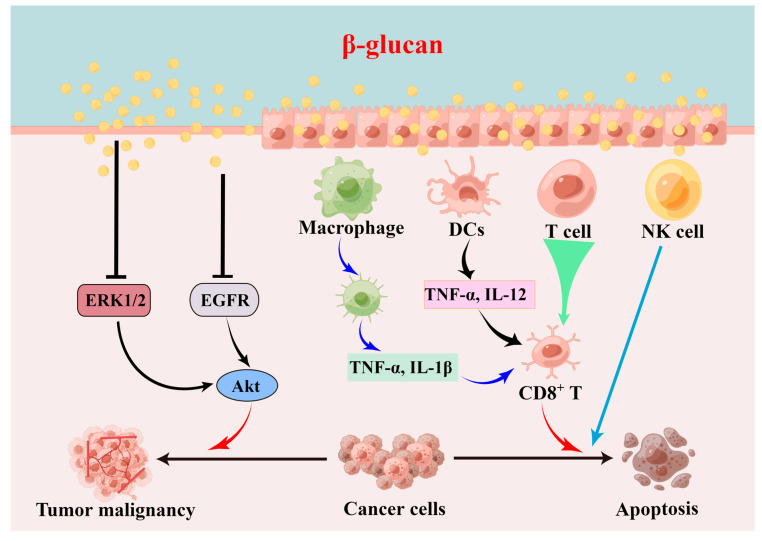
Mechanisms of the antitumor activity of β-d-glucan. This figure is drawn with Figdraw (www.figdraw.com, accessed on 14 June 2023).

**Table 1 foods-12-02975-t001:** The structures and bioactivities of β-d-glucan in *Ganoderma* species.

No.	Origin	Extraction	*M_w_*	Structure Features	Ref.
1	Fruiting bodies of *G. lucidum*	Hot water	6.30 × 10^4^	β-(1→6)-D-glucan	[26]
2	Fruiting bodies of *G. lucidum*	Hot water	3.73 × 10^3^	Unknown	[27]
3	Fruiting bodies of *G. lucidum*	Alkali	1.33 × 10^5^	β-(1→3)-D-glucan	[28]
4	Fruiting bodies of *G. lucidum*	Hot water	5.20 × 10^3^	β-(1→3)-D-Glc*p*, β-(1→4)-D-Glc*p*, β-(1→6)-D-Glc*p* linked residues substituted at O-6 with (1→3)-D-Glc*p* and (1→4)-D-Glc*p*	[29]
5	Fruiting bodies of *G. lucidum*	Alkali	3.98 × 10^3^	Unknown	[30]
6	Fruiting bodies of *G. lucidum*	Hot water	3.75 × 10^6^	β-(1→3)-D-glucan with a (1→6)-D-Glc*p* side-branching unit on every third residue	[31]
7	Fruiting bodies of *G. lucidum*	Alkali	1.33 × 10^5^	β-(1→3)-D-glucan with few branches at C-6 and C-2 positions	[7]
8	Fruiting bodies of *G. lucidum*	UAE	1.56 × 10^4^	β-(1→3, 1→6)-D-glucan	[21]
9	Fruiting bodies of *G. lucidum* and *G. sinense*	Hot water	Unknown	β-(1→3)-D-glucan	[32]
10	Fruiting bodies of *G. lucidum*	Hot water	1.95 × 10^4^	→6)-β-d-Glc*p*-(1→ and →3)-β-d-Glc*p*-(1→residues	[24]
11	Fruiting bodies of *G. lucidum*	Hot water	2.07 × 10^4^	β-(1→3) and β-(1→6)-D-glucan	[33]
12	Fruiting bodies of *G. lucidum*	Hot water	2.06 × 10^4^	A mixed (1→3), (1→4), (1→6)-β-d-glucan backbone	[34]
13	Fruiting bodies of *G. lucidum*	Hot water	2.42 × 10^6^	β-(1→3) -D-glucan	[34]
14	Fruiting bodies of *G. atrum*	Hot water	Unknown	→3)-β-d-Glc*p*-(1→3)-β-d-Glc*p*-(1→3)-β-d-Glc*p*-(1→	[35]
15	Fruiting bodies of *G. australe*	Hot water	Unknown	β-(1→3)-D-glucan	[36]
16	Fruiting bodies of *G. sinense*	Hot water	3.20 × 10^4^	β-(1→4)- and (1→6)-D-Glc*p* linked residues substituted at O-3 with (1→6)-D-Glc*p*	[37]
17	Fruiting bodies of *G. resinaceum*	Hot water	2.60 × 10^4^	β-(1→3)-D-glucan	[38]
18	Fruiting bodies of *G. resinaceum*	Alkali	Unknown	A mixed (1→3), (1→4), (1→6)-β-d-glucan backbone	[39]
19	Mycelia of *G. lucidum*	Alkali	9.30 × 10^3^	β-(1→3)-D-glucan	[40]
20	Mycelia of *G. lucidum*	Hot water	10^5^–10^6^	β-(1→3, 1→6)-D-glucan	[41]
21	Mycelia of *G. lucidum*	Hot water	Unknown	β-(1→3)-D-glucan	[42]
22	Spores of *G. lucidum*	Hot water	1.00 × 10^4^	β-(1→3)-D-glucan	[43]
23	Spores of *G. lucidum*	Hot water	1.26 × 10^5^	β-(1→6)-D-glucan	[44]
24	Spores of *G. lucidum*	Hot water	8.00 × 10^3^	Unknown	[45]
25	Spores of *G. lucidum*	Hot water	1.57 × 10^5^	A mixed (1→3), (1→4), (1→6)-β-d-glucan backbone	[46]
26	Spores of *G. lucidum*	Hot water	1.03 × 10^5^	β-(1→6)-D-glucan	[47,48]
27	Spores of *G. lucidum*	Hot water	1.50 × 10^4^	β-(1→3)-D-glucan	[49]
28	Spores of *G. lucidum*	Hot water	1.93 × 10^5^	A mixed (1→3), (1→4), (1→6)-β-d-glucan backbone	[50]
29	Spores of *G. lucidum*	Hot water	8.2 × 10^4^	β-d-Glc*p*-(1→3 or 1→6)-β-d-Glc*p*-(1→	[6]
30	Spores of *G. lucidum*	Hot water	1.28 × 10^5^	β-(1→3)-D-glucan	[51]
31	Fermentation broth of *G. lucidum*	Hot water	Unknown	β-(1→3)-D-glucan	[52]

Note: UAE: ultrasonic-associated extraction.

**Table 2 foods-12-02975-t002:** Biological activities of β-d-glucan from *Ganoderma* species.

No.	Resources	Model & Dosage	Effect & Mechanism	Ref.
Immunomodulation				
1	Fruiting bodies of *G. lucidum*	LPS-induced T- and B-lymphocytes 1–100 μg/mL β-d-glucan for 44 h	T- and B-lymphocyte proliferation ↑;	[26]
2	Fruiting bodies of *G. lucidum*	LPS-induced DCs, Th 17 cells 0.01–1 mg/mL β-d-glucan for 12 h LPS-induced CD4^+^ T cells 0.1 mg/mL β-d-glucan for 60 h	Levels of TNF-α, IL-23, IL-10 ↑and IL-6 ↓; CD4^+^ T cell proliferation ↑and IL-17 and IL-4 levels ↑; MEK-ERK pathway ↑;	[58]
3	Fruiting bodies of *G. lucidum*	LPS-induced macrophages RAW264.7 50–500 µg/mL β-d-glucan for 48 h	NO production ↑;	[31]
4	Fruiting bodies of *G. lucidum*	LPS-induced macrophages RAW264.7 5–100 µg/mL β-d-glucan for 24 h	NO production and TNF-α level ↓ NF-κB-JNK1/2-ERK1/2 pathway ↓;	[7]
5	Fruiting bodies of *G. lucidum*	CTX-induced immunosuppression mice model 250 mg/kg/BW β-d-glucan for 7 days	Numbers of WBC and LC ↓; Levels of IgA and IgG ↑;	[24]
6	Fruiting bodies of *G. lucidum*	hDectin-1a cells 10–200 μg/mL β-d-glucan for 24 h	Expression of SEAP ↑; NF-κB pathway ↑;	[34]
7	Fruiting bodies of *G. lucidum*	LPS-induced macrophages RAW264.7 300 μg/mL β-d-glucan for 12, 24, 48 h	Expression of IL-1β and IL-6 ↓; Expression of TLR2, TLR4, TLR6, and iNOS ↑; NF-κB pathway ↑;	[59]
8	Fruiting bodies of *G. sinense*	PMB-induced mouse splenic B cells 30 µg/mL β-d-glucan for 24 h LPS-induced macrophages RAW264.7 100–800 µg/mL β-d-glucan for 24 h LPS-induced human PBMCs and moDCs 0.3–1000 ng/mL β-d-glucan for 18 h (for PBMCs) and 48 (for moDCs)	Mouse splenic B cell proliferation and NO production ↑; Levels of IL-1β, TNF-α, IL-10 and IL-12p40 ↑;	[37]
9	Fruiting bodies of *G. australe*	LPS-induced macrophage 0–2.5 μg/mL β-d-glucan for 48 h	Level of IL-6 and phagocytic activity ↑;	[36]
10	Mycelia of *G. lucidum*	Mice treated with inactive *Micrococcus lysodeikticus* 8 mg/mL β-d-glucan for 1 weeks	Levels of IgA or IgG ↑; Poly-Ig receptor expression ↑;	[60]
11	Mycelia of *G. lucidum*	LPS-induced RAW264.7 cells 0–100 μg/mL β-d-glucan for 48 h	Levels of TNF-α ↑;	[41]
12	Spores of *G. lucidum*	LPS-induced lymphocytes 25 mg/kg/BW β-d-glucan for 4 days	T and B cell proliferation ↓; Antibody production ↓;	[44]
13	Spores of *G. lucidum*	DNCB-induced delayed-type ear swelling mice model 75–300 mg/kg/BW β-d-glucan for 7 days	Ear swelling ↑;	[46]
14	Spores of *G. lucidum*	CTX-induced immunosuppression mice model 300 mg/kg/BW β-d-glucan for 7 days	HC_50_ ↑;	[50]
15	Spores of *G. lucidum*	RAW264.7 cells 5–200 μg/mL β-d-glucan for 24 h	Macrophage activation ↑; NO production ↑;	[51]
Anti-inflammatory				
1	Fruiting bodies of *G. lucidum*	Caco-2 cells induced by LPS 10–200 μg/mL β-d-glucan for 2 h DSS-induced colitis mice model 10–200 mg/kg/BW β-d-glucan for 16 days	Levels of TNF-α, IL-8, MIF and MCP-1 ↓; Colon length of mice ↑; Levels of IL-1β and IL-6 ↓;	[56]
2	Spores of *G. lucidum*	DSS-induced colitis 393.75 g/kg/BW β-d-glucan for 3 weeks	BW and levels of acetic acid, propionic acid, butyric acid, and total SCFA ↑; SCFA-producing bacteria *Ruminococcus_1* numbers ↑; *Escherichia-Shigella* numbers ↓;	[48]
3	Spores of *G. lucidum*	IEC-6 cells 10–200 μg/mL β-d-glucan for 24 h	IEC-6 cell proliferation and NO production ↑; Expression of IL-1β and IL-6 ↓;	[6]
Antitumor				
1	Fruiting bodies of *G. lucidum*	LLC mice model 25 and 100 mg/kg/BW β-d-glucan for 14 days	Tumor growth and weight ↓; IFN-γ and IL-12 ↑; CARD9/NF-κB/IDO pathway ↑;	[61]
2	Fruiting bodies of *G. lucidum*	Jurkat cells 25 and 50 mg/L β-d-glucan for 48 h	Cell apoptosis ↑; Expression of Bax and caspase-3 ↑and Bcl-2 ↓;	[33]
3	Mycelia of *G. lucidum*	LLC mice model 80% β-d-glucan for 29 days	Tumor growth and metastasis ↓; Survival time ↑;	[62]
4	Spores of *G. lucidum*	Resident murine peritoneal macrophages 100 μg/mL β-d-glucan for 24 h LLC mice model 50–200 mg/kg/BW β-d-glucan for 10 days	TNF-α and IL-6 ↑; Tumor weight ↓; ERK1/2-MAPK pathway ↑;	[45]
5	Spores of *G. lucidum*	S180-bearing mice model 3–100 mg/kg/BW β-d-glucan for 21 days	Tumor growth ↓;	[49]
6	Fermentation broth of *G. lucidum*	ORL-48 cells 0–4.0 mg/mL β-d-glucan for 72 h	IC_50_ value of cell inhibition was 0.23 mg/mL;	[52]
7	Fermentation broth of *G. formosanum*	Lung cancer mice model 80 mg/kg/BW β-d-glucan for 32 days	Tumor growth ↓; Proportion of NK cells ↑; Foxp3, IL-10, and Notch1 expression ↓;	[63]
Antioxidant				
1	Fruiting bodies of *G. lucidum*	Hydroxyl radical 0.16–10 mg/mL β-d-glucan Fe^2+^-chelating activity 1.5–10 mg/mL β-d-glucan	HO^·^ inhibition activity was 78.3% at 1.25 mg/mL Fe^2+^-chelating activity was 58% at 10 mg/mL	[29]
2	Fruiting bodies of *G. lucidum*	H_2_O_2_-induced RAW264.7 cells 0–150 μg/mL β-d-glucan for 24 h	H_2_O_2_-induced macrophage death ↓; ROS production and SMase activity ↓;	[30]
3	Fruiting bodies of *G. lucidum*	DPPH and FRAP 5–100 μg/mL	Highest DPPH scavenging activity was 65.66% Value of FRAP was 0.0036 mmol Fe^2+^/L	[21]
4	Fruiting bodies of *G. lucidum*	DPPH and FRAP 50 mg/mL β-d-glucan	IC_50_ value of DPPH inhibition was 18.34 mg/mL IC_50_ value of FRAP was 18.38 mg/mL	[64]
Others				
1	Fruiting bodies of *G. lucidum*	H_2_O_2_-induced liver injury in HepG2 cells 0.1–0.4 mg/mL β-d-glucan for 24 h Restraint stress-induced live damage mice 100–400 mg/kg/BW β-d-glucan for 7 days	Levels of ALT and AST ↓; MDA content ↓; Levels of GSH-Px, SOD, and CAT ↑;	[65]
2	Fermentation broth and mycelia of *G. lucidum*	STZ-induced diabetic 5 mg/mL β-d-glucan for 2 h	α-glucosidase activity ↓; Fasting blood glucose level ↓;	[66]

Note: BW: body weight; CAT: catalase; CARD9: caspase recruitment domain-containing protein 9; CTX: cyclophosphamide; DCs: Dendritic cells; DNCB: dinitrochlorobenzene; DPPH: 2,2-diphenyl-1-picrylhydrazyl; DSS: dextran sulfate sodium; ERK: extracellular signal-regulated kinase; Foxp3: forkhead box P3; FRAP: ferric reducing antioxidant potential; GSH-Px: glutathione peroxidase; HC_50_: half value of hemolysin; IDO: indoleamine 2,3-dioxygenase; IgA: immunoglobulin A; IFN: interferon; IL: interleukin; iNOS: inducible nitric oxide synthase; LC: lymphocytes cells; LLC: Lewis lung cancer; LPS: lipopolysaccharide; MAPK: mitogen-activated protein kinases; MCP: monocyte chemoattractant protein; MDA: malondialdehyde; MDSCs: myeloid-derived suppressor cells; MEK: MAPK/extracellular signal-regulated kinase; MIF: migration inhibitory factor; NF-κB: nuclear factor kappa-B; NO: nitric oxide; PBMCs: peripheral blood mononuclear cells; PMB: polymixin B; ROS: reactive oxygen species; SCFA: short-chain fatty acid; SEAP: secreted alkaline phosphatase; SOD: superoxide dismutase; STZ: streptozotocin; TNF-α: tumor necrosis factor-α; TLR: toll-like receptors; WBC: white blood cells.

## Data Availability

The data used and analyzed during the current study are available from the corresponding author on academic request (C.T.). The data are not publicly available to preserve the privacy of the data.

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
