# Peer review of "Recent Advances in the Preparation, Structure, and Biological Activities of β-Glucan from Ganoderma Species: A Review"

_foods, 2023, doi:10.3390/foods12152975_

Round 1
Reviewer 1 Report
The review article dealing an important subject and here are my suggestions:
1- Abstract section must be refer to the important of the topic of review.
2- Keywords need to rewrite again.
3- Tables 1 & 2 need to be check from authors.
4- The authors need to make editing of language and grammars.
The authors need to make editing of language and grammars
Author Response
Reviewer(s)' Comments to Author:
1- Abstract section must be refer to the importance of the topic of review.
Response: Thank you very much for your nice suggestion. We have added the importance of the topic of review in the Abstract section.
2- Keywords need to rewrite again.
Response: Thank you very much for your nice suggestion. We have rewritten the Keywords in our revised Manuscript.
3- Tables 1 & 2 need to be check from authors.
Response: Thank you very much for your good comments. We have revised and checked the content of the Tables in our revised manuscript.
4- The authors need to make editing of language and grammars.
Response: We feel sorry for our poor writing, however, we do invite a friend of ours who is a native English speaker from the USA to help polish our article. Due to our friend’s help, the article was edited extensively. And we hope the revised manuscript could be acceptable to you.

Reviewer 2 Report
The objectives of authors in the present manuscript were to collect research in preparation strategies, structural characteristics, biological activities, and mechanisms of Ganoderma β-glucan, and its toxicological assessment and applications. In general, the content was well organized. But there are some suggestions for improving the manuscript are mentioned below:
It is suggested that the authors provide some information about the β-glucan HPLC chromatogram in the section on structural characteristics (2.2).
It is suggested that, if possible, authors add another section titled "Ganoderma genetic studies" to their article. the provide some information about identifying genetic loci and candidate genes related to β-glucan content in Ganoderma?
In the Conclusions and Future Perspectives section, the authors should point out what is the novelty and valuable content in their manuscript in comparison with other similar studies in this field.
Author Response
Reviewer(s)' Comments to Author:
The objectives of authors in the present manuscript were to collect research in preparation strategies, structural characteristics, biological activities, and mechanisms of Ganoderma β-glucan, and its toxicological assessment and applications. In general, the content was well organized. But there are some suggestions for improving the manuscript are mentioned below:
- It is suggested that the authors provide some information about the β-glucan HPLC chromatogram in the section on structural characteristics (2.2).
Response: Thanks for your reminder, we have supplemented the β-glucan HPLC chromatogram in the revised Figure 2.
- It is suggested that, if possible, authors add another section titled "Ganoderma genetic studies" to their article. The provide some information about identifying genetic loci and candidate genes related to β-glucan content in Ganoderma?
Response: Thank you very much for your professional review work on our manuscript. Currently, the structure elucidation and the biosynthetic pathway of β-glucan in edible and medicinal mushrooms are unknown. Of note, the biosynthesis of Ganoderma β-glucan is currently the main interest and ongoing research in our group, and encouraging results have been achieved under several grants (i.e., National Natural Science Foundation of China). Meanwhile, the related research results on the biosynthetic process of β-glucan will be reported and shared in the near future. Therefore, considering that the biosynthesis of β-glucan is still unclear, this review will not discuss the identification information about genetic loci and candidate genes related to β-glucan content in Ganoderma species, but we focus on summarizing the recent research advance in the preparation strategies, structural characteristics, bioactivities, and potential mechanisms of Ganoderma β-glucan, as well as its toxicological assessment and applications. Thanks again for your valuable comments on our manuscript.
- In the Conclusions and Future Perspectives section, the authors should point out what is the novelty and valuable content in their manuscript in comparison with other similar studies in this field.
Response: Thanks for your comments. We have revised the section on Conclusions and Future Perspectives according to your suggestion.

Reviewer 3 Report
There is 31 percent plagiarism. Some of them are technical terms. But its better to go through thoroughly and reduce it

Author Response
Reviewer(s)' Comments to Author:
There is 31 percent plagiarism. Some of them are technical terms. But its better to go through thoroughly and reduce it.
Response: Thanks for your nice suggestion. We have revised and removed the same content as the published article according to the iThenticate report.

Reviewer 4 Report
It is well known that polysaccharides obtained from natural sources have diverse health benefits. Ganoderma is rare medicinal fungi mushroom that has been cultured and consumed worldwide for centuries. as well as has traditionally been used as a remedy for diseases.
Up to now, approximately two hundred polysaccharides have been extracted and identified from the fruiting bodies, mycelia, spores, and fermentation broth of Ganoderma species
Modern pharmacology studies have proved that Ganoderma ß-glucan exhibited versatile biological activities, including immunomodulatory, anti-inflammatory, antitumor, antioxidant, antiviral, and gut microbiota regulation. This means without doubts that ß-D-glucan polysaccharide can be utilized as a health product and for medicinal treatment.
Lately, the extraction, purification, structural characterization, and pharmacological activities of polysaccharides from the fruiting bodies, mycelia, spores, and fermentation broth of Ganoderma species were reported, but thorough studies on the preparation, structure and bioactivity, toxicology and utilization of these ß-D-glucans are nonetheless extremely limited.
In this review the authors present a complete summary of the recent research advance in the preparation strategies, structural characteristics, biological activities, and potential mechanisms of Ganoderma ß-D-glucan polysaccharides. It also deals with its toxicological assessment and applications..
This review is professionally written and provide a critical evaluation of the data available from existing studies. In summary, this review provides useful guidance and reference for the development and application for ß-D-glucan polysaccharides. The only concern of this referee is the bludgeoning of the IUPAC Carbohydrate nomenclature. The authors are encouraged to be consistent with this nomenclature throughout this review.
The following are corrections that should be followed:
1. In page 2, your Abstract , please correct ß-glucan as ß-D-glucan.
2. In page 3, Table 1, please be consistent with The IUPAC Carbohydrate nomenclature:
Foe example : 1,3-linked ß-D-Glcp residues substituted at O-6 with 1,6-linked Glcp residues
Should be written as: (1→3)- ß-D-Glcp linked residues substituted at O-6 with (1→6)-D- Glcp residues.
D must be one font lower than the text and p should be in italics
3. In page 5, line 113-115, please correct as shown:
The analysis of structural features indicated that ß-D-glucan from Ganoderma species was a liner polymer composed of glucose molecules connected by a ß-D-(1→3), -(1→4), and -(1→6)-linked main chain and ß-D-(1→6)-linked branches [10, 28, 29]
4. In page 5, Figure 2, your second drawing of this table is wrong! Misses one hydroxyl group
A well written reveiw
Author Response
Reviewer(s)' Comments to Author:
The following are corrections that should be followed:
- In page 2, your Abstract, please correct β-glucan as β-D-glucan.
Response: Thanks for your nice suggestion, we have corrected β-glucan as β-D-glucan in our revised manuscript.
- In page 3, Table 1, please be consistent with The IUPAC Carbohydrate nomenclature:
For example: 1,3-linked β-D-Glcp residues substituted at O-6 with 1,6-linked Glcp residues
Should be written as: (1→3)-β-D-Glcp linked residues substituted at O-6 with (1→6)-D-Glcp residues. D must be one font lower than the text and p should be in italics
Response: Thanks for your nice suggestion, we have modified and marked with red color in our revised manuscript.
- In page 5, line 113-115, please correct as shown:
The analysis of structural features indicated that β-D-glucan from Ganoderma species was a liner polymer composed of glucose molecules connected by a β-D-(1→3), -(1→4), and -(1→6)-linked main chain and β-D-(1→6)-linked branches [10, 28, 29]
Response: Thanks for your nice suggestion, we have corrected it in our revised manuscript.
- In page 5, Figure 2, your second drawing of this table is wrong! Misses one hydroxyl group
Response: Thanks for your reminder. We have revised the above error in our revised manuscript.
